# Leveraging hillslope connectivity for improved large-scale assessments of landslide risk

- Rafael J. P. Schmitt<sup># 1,2</sup>, Shikshita Bhandari<sup># 3</sup>, Adrian Vogl <sup>2</sup>, Odin Marc <sup>4</sup>
  - <sup>1</sup> Environmental Studies Program, University of California Santa Barbara, Santa Barbara, CA
  - <sup>2</sup> The Natural Capital Project, Stanford University, Stanford, CA
  - <sup>3</sup> Earth System Science, Stanford University, Stanford, CA
  - <sup>4</sup> Université de Toulouse, CNRS, IRD, CNES, GET, Toulouse, France
- Correspondence to: Rafael J. P. Schmitt (<u>rschmitt@ucsb.edu</u>), Shikshita Bhandari (shikshitabhandari@stanford.edu)
  - # Contributed equally

## 1. Abstract

Landslides hinder sustainable development in mountain regions, threatening livelihoods and impacting linear and water infrastructure. Susceptibility maps are a common tool for estimating and managing landslide hazards, exposure, and risks. Yet, susceptibility maps omit hillslope connectivity, a critical shortcoming for mapping the magnitude of landslide hazards, including cascading hazards from slope failure and downslope runout. Herein we propose the COHESION (COnnected HillslopE Susceptibility for slOpefailure and ruNout) approach to couple susceptibility mapping with an assessment of hillslope connectivity to identify downslope-connected landslide objects (LSOs) and associated runout pathways. As we demonstrate for the Kaligandaki basin in Nepal, analyzing LSOs enables estimating the magnitude of slope failures in terms of mobilized sediment volume and to quantify additional impacts from landslide runout. After calibration using a remotely sensed landslide inventory, we find that 16 % of the basin's slopes are susceptible to failure, while an additional 9 % of the basin area is impacted by runout. Around 33 % of buildings and 65 % of roads in the basin are on susceptible slopes, while more than 27 % of buildings and 69 % of roads are in landslide runout pathways. Omitting runout from landslide assessments would thus result in a major underestimation of risk. Our results emphasize the importance of connectivity for slope stability modeling on landscape scales, leading to improved assessments of slope hazards and management of river basin sediments.

# 2. Introduction

Like through a magnifying glass, the world's mountain regions epitomize challenges for sustainable human development (Grumbine and Xu, 2021). Mountain ranges cover 13 % of the worlds surface, are home to a large and growing human population (Thornton et al., 2022), support many more through their role as "water towers of the world" (Immerzeel et al., 2020), and harbor outstanding biodiversity (Ly et al., 2023). Yet, human development in mountain regions lags lowlands.

Landslides are a significant hindrance for human development and reflect many of the unique development challenges of mountain regions (Emberson et al., 2022; Marc et al., 2023). While landslides are a natural part of the evolution of mountain landscapes, they turn into significant natural hazards when they intersect with human populations. Worldwide, it has been estimated that at least 4600 people per year lost their lives in landslides between 2004 and 2010 (Petley, 2012). Even when no lives are lost, landslides are highly disruptive for socio-economic activities and human development. Landslides destroy croplands and impact infrastructure. Notably, roads (Mey et al., 2023; Meyer et al., 2015) and other linear infrastructure such as power lines are vulnerable to landslides (Emberson et al., 2020), impeding access to education, markets, medical care, and energy security. As landslides contribute sediment to rivers, their occurrence impacts downstream water infrastructure, such as dams and hydroelectric powerplants (Fort et al., 2010; Schwanghart et al., 2018), with repercussion for sectors such as irrigation and hydropower.

Slope failures and the resulting landslides are natural processes, but human activities increase their occurrence across scales. On global scales, a warming climate increases the probability of extreme precipitation events that can trigger shallow landslides (Gariano and Guzzetti, 2016). On regional scales deforestation and climate-driven changes in vegetation can increase landslide occurrence, as vegetation support soils with their root system (McGuire et al., 2016; Sidle et al., 2006a, b; Swanson and Dyrness, 1975). On local scales, road construction is notorious for causing landslides because of road impacts on slope hydrology and statics (McAdoo et al., 2018; Sudmeier-Rieux et al., 2019; Vuillez et al., 2018). Landslide risks for roads are thus a prime example for how humans are both impacted by, and drivers behind, increases in natural hazards and associated risks.

Managing the impacts of landslides is best addressed by a two-pronged approach. On the one hand, better planning can reduce exposure, for example by avoiding landslide-prone zones when developing infrastructure or settlements. On the other hand, there are opportunities for reducing hazards. For instance, roads can be constructed with better drainage and cut-and-fill slopes can be stabilized with grey engineering or nature-based solutions. Proactive land use policies can help to encourage reforestation in landslide-prone areas. Such nature-based solutions can create co-benefits for people and nature, and the reduction of landslide hazards often motivates sustainable land use planning in mountain regions (Sidle et al., 2006b; Vogl et al., 2019a). Understanding landslide hazards and exposure is thus of relevance for disaster awareness and prevention and is critical for sustainable human development in the world's mountain regions.

The numerical modeling of landslides has a long legacy, yet significant gaps remain in bringing the conceptual understanding of hillslope processes into landscape-scale slope stability assessments. Such assessments would be critical for landslide management and for understanding the contribution of mountain ecosystems to reducing landslide hazards. Finite elements and similar methods are now commonly used to study slope stability for individual slopes (Holcombe et al., 2012; Thiebes et al., 2014; Wilkinson et al., 2002). Despite numerical advances, such approaches remain computationally very demanding and require a large amount of data regarding sub-soil properties and slope hydrology, which are challenging to collect even for a single slope (see, e.g., reviews in (Intrieri et al., 2019; Jiang et al., 2022)). Thus, those approaches are not suitable to inform larger-scale studies of hazards, risks and adaptation opportunities, particularly in data-scarce regions. For regional scales, susceptibility mapping approaches abound, as evidenced by the fact that a search on Web of Science revealed 150 review articles for "landslide susceptibility" since 2010. Broadly, susceptibility mapping approaches fall into two classes. Firstly, statistical methods, including machine learning (Huang and Zhao, 2018; Zhang et al., 2023), can be used to link observed slope failures to environmental covariates (e.g., soil characteristics, landuse legacy, and precipitation) and to extrapolate observations to larger scales. Secondly, "factor of safety" (FOS) models are based on the Mohr Coulomb failure criterion for infinite slopes as a measure landslide susceptibility that can be readily calculated with globally available data (Dietrich et al., 1995; Montgomery David R. and Dietrich William E., 2010; Vanacker et al., 2003).

100

Common susceptibility mapping approaches lack consideration of hillslope connectivity, a significant shortcoming, which impedes the understanding and management of landslide risk. Typically, susceptibility maps are derived in a gridded model domain ("rasters") consisting of individual grid cells. Susceptibility values are then determined for each of the individual cells. The resulting maps represent the distribution of landslide hazards in a spatially continuous grid, but do not consider the connectivity between adjacent grid cells. This seemingly minor technical distinction has major implications for management and science.

Landscapes are highly connected systems (Heckmann et al., 2015; Heckmann and Schwanghart, 2013). This is obvious for landslides, where the failure of one part of a hillslope (represented as a cell) could trigger the failure of surrounding cells. Without considering connectivity, the ability to estimate two parameters that are critical for estimating landslide hazards remains limited. Firstly, the mass mobilized in a landslide has been shown to exponentially increase with the size of the slope failure (Larsen et al., 2010) An increased area of connected slope failures will thus exponentially increase the hazard, and will limit opportunities for mitigation (smaller landslides are more readily addressed with engineered or nature-based solutions). Secondly, runout from an upslope landslide might reach lives and livelihoods even if those locations are not at risk of slope failure. This mechanism of exposure is not considered in common susceptibility mapping approaches, notably because calculating runout pathways would require information on both hillslope connectivity and the initial mass of a landslide (Fan et al., 2017; Rickenmann, 2005)

Herein we set out to develop a connectivity-based approach that is compatible with common approaches for large-scale susceptibility mapping in data-scarce regions. The objective of this work is to demonstrate how concepts of connectivity, now widely used for hillslope (Heckmann et al., 2015; Heckmann and Schwanghart, 2013) and fluvial (Czuba and Foufoula-Georgiou, 2014; Schmitt et al., 2016) processes, can be applied to landslide assessments. We demonstrate of application for a major river basin in Nepal, because Nepal is amongst the countries where landslides are most destructive, with 78 fatalities per year reported on average from 1978-2005 (Petley, 2012).

Our framework is based on globally available data, except for remote sensing data on observed landslides (Marc et al., 2018, 2019a) which we use for model verification. In a nutshell, our framework uses single-cell probabilistic approach to slope stability (how likely a cell is to fail given the observed precipitation regimes) similar to other susceptibility mapping approaches (Vanacker et al., 2003). We then use geomorphic terrain analysis (Schwanghart and Kuhn, 2010) to group susceptible cells into downslope-connected groups, or landslide objects (LSOs) a term we borrow from image classification for landslide detection (Ghorbanzadeh et al., 2022)). For each LSO, we calculate magnitude (i.e., volume and mass), failure probability, and delineate the runout pathway. Finally, we overlay information on buildings and roads, derived from Open Street Maps (OpenStreetMap contributors, 2017), to estimate exposure and risk for people and livelihoods.

We propose that combining susceptibility maps with connectivity-based assessments is a relatively straightforward and scalable way to improve the usefulness of susceptibility mapping, with applications ranging from creating disaster awareness, studying future risks under different landuse and climate scenarios to designing nature-based solutions for landslide control. These insights can be useful for managing landslide risk and the associated impacts on human livelihoods in mountain regions around the world.

## 3. Study Area

Our study area, the Kaligandaki basin, lies in the Nepal Himalayas with elevation ranging from 504 m to 8144 m. Rainfall mostly occurs in the monsoon season (June-September) with average of about 500 mm/year near Tibetan plateau to 2000 mm/year in Himalayas (Struck et al., 2015). The 7600 km² expanse

of the Kaligandaki Basin spans seven districts of central Nepal namely Mustang, Myagdi, Baglung, Parbat, Syangja, Kaski and Gulmi with total population exceeding half a million (Nepal Population and Housing Census 2021). Total road network length, as mapped from OpenStreetMap is 6093 km (OpenStreetMap contributors, 2017) with additional roads under construction. There are 54 present and planned hydropower projects in this region with a total capacity of 1853 MW, of which hydropower projects producing a total of 172 MW are currently operational (Nepal Hydropower Portal, 2025).

Figure 1: Map of the study area, comprising the Kaligandaki Watershed in Nepal. A: elevation and general location of the Kaligandaki basin. Panels B and C show available information on building footprints (B) and roads (B). Road and building footprint layers © OpenStreetMap contributors 2025. Distributed under the Open Data Commons Open Database License (ODbL) v1.0.

Since 2010, the Kaligandaki region alone has experienced over 400 larger landslides, resulting in 206 deaths, the destruction of 201 houses, and infrastructure damages totaling around 57.5 million Nepalese Rupees, around 400,000 USD (National Disaster Risk Reduction and Management Authority, 2025). However, national databases might significantly underestimate both the frequency of landslide incidents and the associated losses. For instance, Marc et al. (2019) recorded over 1,000 landslides in a smaller area within the Kaligandaki region from 2010 to 2015, which contrasts sharply with official records. Because of the fragile geology, triggering factors like earthquakes and heavy rainfall, and anthropogenic activities like road construction, the number of landslide incidents has been escalating in recent years in Nepal (Kc et al., 2024; McAdoo et al., 2018).

#### 4. Methods

Herein we introduce an approach to evaluate slope susceptibility based on the Mohr Coulomb failure criterion, and stochastic analysis of precipitation events (sections 4.1 - 4.3). We then couple the resulting maps of pixel-level susceptibility with an assessment of hillslope connectivity, resulting in the COHESION (**CO**nnected **H**illslope Susceptibility for sl**O**pefailure and ru**N**out) model (section 4.4), which builds on ideas laid out by some of us in an earlier report (Vogl et al., 2019a, b; World Bank., 2019). The COHESION model translates pixel-level maps of slope susceptibility into information on the size of a slope failure (section 4.5), the resulting downslope runout (section 4.6). This information can finally be used to assess risks to infrastructure (section 4.7). Finally, we present how we calibrated and benchmarked the model with remotely sensed landslides from the Kaligandaki watershed from Marc et al. (2019a), Figure 2).

Figure 2: Combining susceptibility mapping and hillslope connectivity into a framework for evaluating landslide risk (COHESION: COnnected Hillslope Susceptibility for slOpefailure and ruNout). Boxes detail the workflow and the section in which the methods are described. The result is a wider perspective on landslide risks, including landslide

190

200

205

magnitude and runout, than what would be possible by susceptibility mapping approaches that do not consider connectivity. Note that different methods can be used for pixel-level susceptibility assessments.

## 4.1. Factor of safety (FOS) based susceptibility mapping

At large scales, landslide susceptibility is often evaluated by assessing the stability of individual grid cells within a digital elevation model (DEM) using the Mohr–Coulomb failure criterion (Dietrich et al., 1995; Huang and Zhao, 2018; Montgomery David R. and Dietrich William E., 2010; Vanacker et al., 2003). Under the assumption of an infinite slope with a shallow, planar failure surface parallel to the ground, slope stability is measured through the factor of safety, defined as the ratio between the available shear strength,  $\tau_i$ , and shear stress,  $\tau_{m,i}$ , (where i denotes a cell in a DEM)

$$FS_i = \frac{\tau_i}{\tau_{mi}}$$

A cell is considered susceptible to failure when its factor of safety falls below one ( $FS_i 

$$m = \frac{q_i * A_{D,i}}{B_i * T_i}$$

(O'Loughlin, 1986).  $q_i$  is the specific subsurface flow into i [m/s].  $A_{D,i}$  is the contributing upslope area [m<sup>2</sup>],  $B_i$  denotes the side length of i [m] which equals the DEM resolution, while the transmissivity  $T_i$  is defined as

$$T_i = K * z_i * cos\alpha$$

4

[m²/s], where *K* is the hydraulic conductivity [m/s]. Conceptually, equation 3 compares the incoming subsurface water from the contributing area to the soil's ability to conduct water at the downslope edge of the cell.

In equation 3,  $q_i$  can be derived from a water balance considering for upslope precipitation p minus evapotranspiration (ET) and surface runoff ( $Q_R$ ) (all in meters)

$$q_i * A_{D,i} = Q_i = \sum_{j \in AD_i} (p_j - ET_j - Q_{R,j} - Q_{D,j}) * B_j^2$$

5

Herein, we estimate evapotranspiration from gridded global reference evaporation  $(ET_0)$  and a landcover coefficient,  $k_f$ , so that

$$ET_i = ET0_i * kf_i$$

6

Lastly,  $Q_{R,j}$ ,  $Q_{D,j}$  and denote surface runoff, and deep percolation, i.e., water that is not contributing to soil water saturation. While we do not account for deep percolation,. We use Kent's (1973) curve number model to estimate surface

$$Q_{R,i} = \begin{cases} \frac{(p_i - 0.2S_i)^2}{p_{ij} + 0.8S_i} & \text{if } I_{a,i} 

 $CN_j$  reflects local land use and soil properties and is documented extensively in standard engineering references and textbooks (Rawls et al., 1992)..

#### 4.3. Factor of safety calculations under consideration of rainfall probabilities

The values of many parameters of the factor of safety equation (equation 1), such as slope or soil depth, vary in space, but not in time over management-relevant timespans. In contrast, soil moisture fluctuates not only in space, but also in time in response to precipitation events. Herein, we consider for the spatiotemporal variability in soil moisture through a statistical approach. Specifically, we solve equation 1 for  $m_i$ 

245

255

$$m_i = \left[ \frac{FS_i * \tau_{m,i} - (c_i + \delta c_i)}{z_i * \cos^2 \alpha_i * \tan \phi_i} - \gamma_{s,i} \right] \frac{1}{-\gamma_w}$$

10

11

with that we can solve for condition of  $FS_i \le 1$ , i.e., when slope failure will occur. Equation 10 can then be used to define a threshold saturation,  $m_i^*$ , at a value of  $FS_i = 1$ 

$$m_i^* = \left[\frac{1 * \tau_{m,i} - (c_i + \delta c_i)}{z_i * \cos^2 \alpha_i * \tan \phi_i} - \gamma_{s,i}\right] \frac{1}{-\gamma_w}$$

Based on this, we then substitute  $m_i^*$  into equation 3

$$m_i^* = \frac{Q_i^*}{b_i T_i}$$

12

to define a threshold subsurface flow

$$Q_i^* m_i^* b_i T_i = Q_i^*$$

13

(Vogl et al., 2019a, b; World Bank., 2019).

Given that landslides in subtropical settings are generally triggered in the wet season (Dahal and Hasegawa, 2008) we define  $Q_i^*$  based on average wet season conditions as

$$Q_i^* = \overline{Q}_i + Q_i(e)^*$$

14

Here,  $Q_i^*$  is the threshold subsurface flow,  $\overline{Q}_i$  is the average antecedent subsurface flow during the wet season, and  $Q_i(e)$  is the additional subsurface flow from a rain event e.

We then use the curve number approach and a water balance for cell i to determine the threshold rainfall that would be required to reach the threshold soil moisture conditions during an event e, under consideration of average antecedent moisture conditions during an average wet season.

$$Q_i^*(e) = p_i^*(e) - ET_i(e) - Q_{R,i}(e)$$

270

Thus,

$$p_i^*(e) = Q_i^*(e) + ET_i(e) + Q_{R,i}(e)$$

can be interpreted as the critical precipitation amount needed to trigger failure at cell i, given typical moisture conditions during an average wet season and the hydrologic partitioning of rainfall. Evaluating the likelihood of  $p_i^*(e)$  being exceeded then depends on local rainfall characteristics. We assume that the annual precipitation maxima at cell i follow an extreme value distribution described as:

$$f(p_i|\mu_i,\sigma_i) = EV(p_i) = \frac{1}{\sigma_i} e^{\frac{p_i - \mu_i}{\sigma_i}} e^{-exp\left(\left(\frac{p_i - \mu_i}{\sigma_i}\right)\right)}$$

16

 $f(p_i | \mu_i, \sigma_i)$  denotes the probability of an event with precipitation  $p, \mu_i$  and  $\sigma_i$  are the scale and location parameters of annual rainfall maxima. The corresponding cumulative distribution of  $p_i$  is

$$F(p_i|\mu_i,\sigma_i) = \frac{1}{\sigma_i} exp\left(-exp\left(\frac{\mu_i - p_i}{\sigma_i}\right)\right)$$

17

Thus  $F(p 

310

315

300 slope failure, expressed as the likelihood that conditions at cell *i* roduce a factor of safety below one.

Figure 3: Extreme value distribution parameters (equation 21) for annual precipitation maxima across the Kaligandaki catchment, derived from gauge records (white squares) and interpolated to a continuous surface using kriging. This spatial dataset enables estimating the probability of exceedance for any rainfall threshold at the cell level. Connected slope failure assessments

Slope failures are unlikely to be confined within the arbitrary limits of single raster cells. Understanding this spatial linkage is crucial for three reasons. (i) Because landslide volume increases nonlinearly with area, it is important to delineate contiguous clusters of unstable cells rather than treat each pixel in isolation (Larsen et al., 2010). (ii) Larger volumes increase runout distances, influencing how far debris will travel and what downslope assets may be affected (Rickenmann, 2005). (iii) Connectivity also determines whether mobilized material enters the fluvial sediment system, with implications for catchment sediment budgets and downstream infrastructure (Dow et al., 2024). Traditional pixel-based hazard maps cannot provide this information. We therefore present a method that groups individual, potentially unstable, cells into downslope-connected landslide objects (LSOs), and show how delineating LSOs enables estimating landslide volume and to delineating runout-prone zones.

To derive LSOs, we firstly perform a classification of all cells according to their potential failure probability. The lowest risk of failure is when saturation is zero, i.e.,

$$min(FS_i) = FS(m_i = 0)$$

On the opposite, the highest risk of failure is when soils are fully saturated

$$\max(FS_i) = FS(m_i = 1)$$

Based on these criteria, each slope cell can be classified into one of three conditions:

- I)  $FS(m_i = 0) < 1$ . These cells are unstable even when completely dry and therefore represent areas likely lacking a stable soil mantle.
- 2)  $FS(m_i = 1) > 1$ . These cells remain stable even under fully saturated conditions.
- 325 3)  $FS(m_i^*) \le 1$ . These cells can become unstable only under certain soil moisture conditions.

We filter out all cells for conditions (1) and (2) because slope failure cannot occur for those cells. Instead, cells with condition (3) are potentially susceptible to failure (Figure 4 a, red cells). To delineate contiguous zones of instability, we then group conditionally unstable cells that are connected along the downslope flow direction using functions from TopoToolbox (Schwanghart and Kuhn, 2010).

Each connected group of cells is treated as a single landslide object (LSO) and assigned a unique identifier k. A cell  $i \in LSO_k$  belongs to the landslide object k, which consists of  $n_k$  cells. We then derive the properties of each  $LSO_k$  based on the characteristics of its constituent cells.

The next step is to estimate the joint probability that all cells within  $LSO_k$  will fail. Because these cells are physically connected and exposed to similar rainfall forcing, their failure probabilities cannot be considered independent. We therefore assume that the overall failure probability of  $LSO_k$  can be expressed as:

$$F(k) = \frac{\sum_{i \in k} (1 - F(p_i | \mu_i, \sigma_i))}{n_k}$$

19

i.e., that the failure probability of  $LSO_k$  is the average of failure probabilities for all cells belonging to  $LSO_k$ . This assumption could be replaced with alternatives, e.g.,

$$F(k) = \min_{i \in k} \left( 1 - F(p_i | \mu_i, \sigma_i) \right)$$

his corresponds to a "weakest-link" assumption, meaning that failure of the entire slope is initiated once the most vulnerable cell fails.

Figure 4: Workflow for delineating connected landslide objects (LSOs) and estimating their failure probabilities. (a) Conditionally unstable cells are identified. (b) These cells are grouped into downslope-connected LSOs (shown in different colors). (c) Cell-scale failure probabilities (blue) are aggregated to the LSO level (green to red outlines).

#### 4.4. Estimating LSO volume

Empirical studies show that landslide volume scales nonlinearly with landslide size. Using a global dataset of landslide scars, Larsen et al. (2010) derived a power-law relationship linking landslide surface area to the volume of mobilized sediment as:

$$V_{LS} = \alpha * A_{LS}^{\gamma}$$

21

with  $V_{LS}$  and  $A_{LS}$  being the volume  $[m^3]$  and the area  $[m^2]$  of a landslide. Based on a set of more than 4000 observations, they found a best fit between Equation 21 with  $\log \alpha = 0.86$  (i.e.,  $\alpha = 10^{0.86} = 7.24$ ) and  $\gamma = 1.322$  with an  $R^2 = 0.95$ .

Using these definitions, the area of an LSO can be computed from the areas of its constituent cells

$$\mathbf{A}_{\mathrm{LSO,k}} = \sum\nolimits_{i \in k} {{B^2}}$$

360

Where b is the side length of a cell in the DEM so that the volume of a LSO  $[m^3]$  finally reads as

$$V_{LSO,k} = 7.24 * A_{LSO,k}^{1.322}$$

23

## 365 4.5. Calculating landslide runout

As a final step, we relate landslide runout length to the volume of mobilized material. The distance traveled by debris is controlled both by landslide volume, which determines the available kinetic energy, and by slope gradient. Consequently, large failures on steep slopes generally produce longer runouts than small failures on gentle slopes. This relationship has been established through empirical studies (Rickenmann, 1999, 2005). Following the equations presented in Rickenmann (1999, 2005) we calculate runout downslope of an LSO as

$$L_{LSO} = 1.9V_{LSO}^{0.16}H_{LSO}^{0.83}$$

24

L<sub>LSO</sub> is the runout length and H<sub>LSO</sub> is the vertical drop between the starting point of runout (the lowest point of the landslide scar) and the downslope end point along the runout path (both in meters). These two quantities are not independent, as longer runout distances naturally involve greater vertical drops. In a gridded model, it is practical to evaluate the condition in Equations 30 sequentially for all cells along each potential downslope path. For every step downslope, both cumulative travel distance and vertical drop are updated. Runout is considered to terminate once the predicted runout length from Equations 30 becomes shorter than the remaining distance to the next downslope cell or as soon as the runout encounters a river.

To perform this calculation, we first delineate all cells located between  $LSO_k$  and the channel network, denoting this downslope runout path as  $\gamma_k$ . For any cell  $i \in \gamma_k$ ,  $\delta H_{hi}$  represents the elevation drop from  $LSO_k$  to i and  $\delta L_h$  is the horizontal distance from  $LSO_k$  to i. Moving along all cells in  $\gamma$ , we can then compute the cumulative L and H ratio at each cell (Figure 5).

405

Figure 5: Schematic of runout modeling in a gridded domain. Individual cells may lie on the runout paths ( $\gamma$ \gamma $\gamma$ ) of multiple landslides. The inset shows a longitudinal hillslope profile (A–B) illustrating how runout length is estimated using the empirical relationship of Rickenmann (2005).

## 4.6. Calculation of risks to structure and roads

When assessing risks from landslides, it is important to distinguish between calculating hazard for assets located on an LSO and for assets located on its runout path. Any given asset, such as a building or a road, can be only on a single LSO. Instead, runout paths of several different LSOs can overlap (Figure 5). For instance, suppose cell *i* lies on the runout pathways of LSOs *l*, *m*, and *k*. For this paper we adopt a "worst case" approach

$$F(h) = \max_{F}(k, l, m)$$

in which the runout hazard of a cell is defined by the maximum failure probability F for any upslope LSO.

#### 4.7. Model verification and calibration

Model verification and calibration was performed using the database of landslides from Marc et al. (2019) in the central part of the Kaligandaki area. This database consists of 1170 remotely sensed landslide scars observed in the period from April 2010 to October 2015.

Direct comparison of the observational data and model results is challenging, because observational data capture only a brief snapshot of slope failures while our model represents long-term average failure probabilities, with probabilities estimated from 20 year long time series of precipitation (1985-2005 for most stations). We thus resort to comparing the distribution of failed slopes between model and observations. By doing so, we aim to measure if the model correctly represents the topographic locations

where slopes become unstable under the local precipitation regime. For this comparison, we firstly calculate the mean slope for all observed landslides and the modelled LSOs from the DEM. We then compare the distribution of observed and modelled slope failures through calculating the Kolmogorov–Smirnov (KS) statistics as a measure for how similar the two distributions are (a KS value of 0 would indicate a perfect match, and a value of 1 would indicate increasing divergence).

To study model sensitivity, we first ran our model by changing various uncertain parameters like soil density  $(\rho_S)$ , soil cohesion (c), internal friction angle  $(\phi)$  and hydraulic conductivity (K). Notably, we focused on subsoil parameters that, while known to be spatially highly variable (Marc et al., 2019b), cannot be derived from remote sensing in a spatially distributed manner. Thus, single values from literature are used in general for larger-scale susceptibility assessments. We found that model results were most sensitive to c and  $\phi$ , which we thus selected for further calibration.

We also found that no combination of c and a single value for  $\phi$  led to satisfactory KS values. We thus adopted a model formulation where  $\phi$  is a function of the local slope angle and global parameter, b. We then used b as parameter for model calibration (see section 4.1 and equation 2).

For calibration, we sampled a parameter space spanning parameter value from 1-10 for c (in steps of 1) and from 0.6 - 1 for b (in steps of 0.05). For each of the resulting 100 combinations we then calculated the KS value by comparing the mean slope of modelled and observed landslides.

## 425 **4.8. Data needs**

430

Table 1 lists the data required for the COHESION framework, as well as specific data sources used for the Kaligandaki watershed. It should be noted that the resolution of the model is determined by the resolution of the digital elevation model. In the case of Kaligandaki, many parameters were derived from global datasets with low resolution (1km at the equator). For those data, we deployed nearest neighbor resampling to match the DEM resolution of 30 m. Table 2 lists a full list of parameters and additional descriptions and references.

Table 1: Data sources used in this paper

| Data                       | Description                                     | Source                                                       |
|----------------------------|-------------------------------------------------|--------------------------------------------------------------|
| Digital Elevation<br>Model | ASTER, 30m GeoTIFF                              | ASTER (Earth Science Data Systems, 2025)                     |
| Land Use Land Cover        | National Land Cover/Land Use, 2000, 30m GeoTIFF | Nepal Department of Survey, 2000                             |
| Soil Depth                 | 1km GeoTIFF                                     | ISRIC Soil Grids:<br>https://www.isric.org/explore/soilgrids |
| Precipitation              | Gauged precipitation data (1985-2005)           | Department of Hydrology and<br>Meteorology, Nepal            |
| Building Footprints        | Polygon shapefiles                              | Open Street Map (OpenStreetMap contributors, 2017)           |
| Road Network               | Line and polygon shapefiles                     | Open Street Map (OpenStreetMap contributors, 2017)           |

https://doi.org/10.5194/egusphere-2025-3733 Preprint. Discussion started: 15 October 2025 © Author(s) 2025. CC BY 4.0 License.

| Observed Landslides | 5m RapidEye Satellite imagery | Marc et al. (2019) |
|---------------------|-------------------------------|--------------------|
|                     |                               |                    |

Table 2: List of parameter symbols, units, names, value sources and used values for this case study. Note that parameters for which the value source is empty (-) are calculated from other input parameters. The column "Global value" lists the numeric values for parameters that are considered constant across the model domain, a "-" in this column indicates that values are spatially distributed.

| Parameter                    | Unit    | Name                                          | Value source                                                | Global Value |
|------------------------------|---------|-----------------------------------------------|-------------------------------------------------------------|--------------|
| $A_{D,i}$                    | $m^2$   | Upslope area                                  | Digital elevation model                                     | _            |
| $A_{LS}$                     | $m^2$   | Landslide area                                | _                                                           | _            |
| $B_i$                        | m       | Cell size                                     | Digital elevation model                                     | _            |
| $I_{a,i}$                    | m/d     | Initial abstraction                           | _                                                           | _            |
| $Q_{D,i}$                    | m³/d    | Subsurface runoff                             | _                                                           | _            |
| $Q_{R,i}$                    | m³/d    | Surface runoff                                | _                                                           | _            |
| $S_i$                        | m       | Maximum soil moisture retention               | -                                                           | _            |
| $T_i$                        | $m^2/s$ | Transmissivity                                | _                                                           | _            |
| $c_i$                        | kPa     | Soil cohesion                                 | Calibration                                                 | _            |
| $m_i$                        | _       | soil water saturation                         | _                                                           | _            |
| $p_i$                        | m/d     | Precipitation                                 | -                                                           | _            |
| $p_i^*$                      | m/d     | Threshold precipitation                       | -                                                           | _            |
| $q_i$                        | $m^3/d$ | Subsurface flow                               | -                                                           | _            |
| $z_i$                        | m       | soil depth                                    | ISRIC soil grids                                            | _            |
| $\alpha_i$                   | _       | Slope angle                                   | Digital elevation model                                     | _            |
| $\gamma_s$                   | kN/m³   | unit weight of soil                           | (Vanacker et al., 2003)                                     | 15.6         |
| $\gamma_w$                   | kN/m³   | unit weight of water                          | (Vanacker et al., 2003)                                     | 9.81         |
| $\delta c_i$                 | kPa     | Root cohesion                                 | (Dhakal and Sidle, 2003; McGuire et al., 2016; Sidle, 1991) | 2            |
| $\mu_i$                      | _       | Location parameter extreme value distribution | Local precipitation data, kriging                           | _            |
| $ ho_{\scriptscriptstyle S}$ | Kg/m³   | Soil density                                  | (Vanacker et al., 2003)                                     | 1600         |
| $\sigma_i$                   | _       | Scale parameter extreme value distribution    | Local precipitation data, kriging                           | _            |
| Н                            | m       | Vertical runout travel distance               | -                                                           | _            |

| k                | m/s   | Saturated hydraulic conductivity                 | Estimated average from Vanacker et al., 2003 | 5e-5 |
|------------------|-------|--------------------------------------------------|----------------------------------------------|------|
| $CN_j$           | _     | Curve number                                     | ISRIC soil grids                             | _    |
| $ET_j$           | m/d   | Actual evapotranspiration                        | -                                            | -    |
| ET0 <sub>i</sub> | m/d   | Reference evapotranspiration                     | WordClim                                     | -    |
| $FS_i$           | _     | Factor of safety                                 | -                                            | _    |
| М                | kg    | Landslide mass                                   | -                                            | _    |
| V                | $m^3$ | Landslide volume                                 | -                                            | _    |
| b                | _     | Scaling parameter $\alpha_i - \phi$ relationship | Calibration                                  | _    |
| φ                | 0     | Internal angle of friction                       | Calibration                                  | -    |

#### 440 **5.** Results

445

450

In this section we provide three core results. Firstly, we describe the results of the model calibration, laying out the results and thus how parameters for further modeling were determined. Secondly, we discuss the patterns and probabilities of modelled landslides. Lastly, we present a high-level scenario analysis, demonstrating how the LSO framework can be used for evaluating changes in landslide occurrence and resulting hazards in response to changes in landscape management.

#### 5.1. Model sensitivity and calibration

Only a small set of parameter values generates model results for which observed and modelled unstable slopes are in good agreement in terms of KS values (Figure 6 a). Specifically, we find that using a global value for  $\phi$ , i.e., the same  $\phi$  everywhere, only results in acceptable KS values (around KS = 0.1) if we select very low values of  $\phi$  ( $\phi$  < 5°) and very high values of c (c>8 kPa) (see dark blue area at bottom of Figure 6 c). We deemed that such low values of  $\phi$  are not in good agreement with observational studies which usually place  $\phi$  between 20° and 50° (Schellart, 2000; Schmidt and Montgomery, 1995).

The alternative model formulation, where

$$\phi = b * \alpha_i$$

result in low KS values for a wider range of parameter values (*b* and *c* in this case). There is some equifinality, as different combinations of b and c lead to similar, low KS values. These parameterization follow a diagonal, with either high values of b and low values of c, or low values of b and high values of c leading to similar results (see blue diagonal band of low KS values in Figure 6 b). Yet, we found that parameterizations at the top left of the parameter space Figure 6 b, e.g., b=0.95 and c = 2 kPa leads to very large fraction of slopes being unstable. We thus selected a value of b=0.65 and c = 5 kPa as final parameterization for the model, a combination that resulted in a low KS values (KS= 0.10). Figure 6 d

shows the resulting cumulative distribution of modelled unstable slopes, compared to the cumulative distribution of slopes for observed slope failures.

Figure 6: Model sensitivity analysis and calibration, by comparing the observed and modelled distributions of unstable slopes. Observations and model results are compared by means of the KS statistic. a) results for models with a single global value of φ, and for models where φ is coupled to α via a constant (b), in both panels, colors show the KS values for different combinations of parameters. (c) and (d) highlight the comparison between observed and modelled cumulative distributions of slopes for two selected points in the parameter space. (d) represents the parameterization used for further experiments.

## 5.2. Patterns of LSO and runout hazards and exposure

Figure 7 shows the spatial distribution of modeled LSOs (A) and runout (B) throughout the Kaligandaki catchment. We find that most LSOs are in the southern part of the basin, below the main range of the Himalayas. As expected, this concurs with locations where steep parts of the basin receive high amounts of rainfall (Figure 3). The general pattern also shows that there are few LSOs in high mountain areas. The reason for this is that most of these very steep slopes are identified as unconditionally unstable and are thus filtered out from the results. The spatial characteristics of LSOs are shown in the cutout in Figure 7 a

https://doi.org/10.5194/egusphere-2025-3733 Preprint. Discussion started: 15 October 2025 © Author(s) 2025. CC BY 4.0 License.

- where an LSO consists of multiple cells that are assigned the same failure probability. We find that the average failure probability of LSOs is 19.4 %/year ( $\sigma$  = 20.7%). The high value of  $\sigma$  indicates that few LSOs with large failure probabilities strongly skew the distribution and summary statistics, when excluding LSOs with a failure probability larger than the 95<sup>th</sup> percentile, we find that average LSO failure probability is around 5.7%/year ( $\sigma$  = 5.7%). The total area of unstable slopes covers 1209 km² or 16 % of the total basin area. After considering the failure probability of each LSO, 0.9% of the study area are expected to become unstable per year.
- Including runout in the analysis greatly increases the area impacted by slope failures. As shown in Figure 7 b, many areas that are below LSOs would be impacted by runout (green in Figure 7 b), and are often in the runout pathways from multiple upslope LSOs. The total area of runout pathways is 626 km² and the total impacted by either slope failures or runouts is 1835 km². Thus, considering only LSOs, without runout, would underestimate the area at risk by 66 %.
- Threshold rainfall intensities are a common metric for landslide occurrence (Gonzalez et al., 2024), and we thus compared COHESION results with observations. For Nepal, previous studies indicated a threshold precipitation of 144 mm/d (Dahal and Hasegawa, 2008), with a range from 132 to 358 mm/d (read from Dahal & Hasegawa, 2008, Figure 6). Our estimates for threshold precipitation are in a similar range with an median threshold precipitation  $p_i^* = 218 \text{ mm/d}$  (see EQ. 18), with percentiles  $p_{25} = 112 \text{ mm/d}$  and  $p_{12} = 216 \text{ mm/d}$  and  $p_{23} = 216 \text{ mm/d}$  (see EQ. 18).
- 112mm/d and p<sub>75</sub>=316mm/d.

Figure 7: The LSO model highlights joint hazards from slope failure and runout on reginal scales. Shown are modelled LSOs (a) and the runout pathways associated with each LSO (b) for the entire Kaligandaki catchment. Magnified cutouts show identical areas between A and B and highlight local patterns of joint hazards that can be derived from the proposed model.

Maps of LSOs, runout pathways, and exposed assets can be used for risk assessments. Figure 8 highlights how the spatially distributed information of the LSO model (i.e., information on natural hazards) can be overlaid with information on exposed assets, such as structures and roads. This information then yields information on risk, i.e., what values are exposed, from which process exposure originates, and with which probability assets might be impacted. Additionally, the LSO model results in information about the magnitude of potential hazards. For instance, hazards can be measured as the probability of an LSO

failing, the size or volume of an LSO, the probability of resulting runoff, or a combination thereof. It should also be noted that the current model does not account for sediment deposition along the runout path, thus an asset located more downslope on runout pathway might be damaged less than a more upslope asset.

Figure 8: The LSO model enables detailed assessments of landslide hazards and exposure. Specifically, the results of the LSO model, which indicate hazards from landslides and runout, can be combined with information on exposed assets, such as structures (magenta) and roads (white lines). Combined, this information enables detailed assessment of what is at risk, from which process, and with which probability. This information would not be available based on common susceptibility analysis. Road and building footprint layers © OpenStreetMap contributors 2025. Distributed under the Open Data Commons Open Database License (ODbL) v1.0.

One opportunity arising from the COHESION model is to perform spatially distributed statistical assessments of landslide hazards for different sectors. For the Kaligandaki area we find that landslide hazards are greater for roads than for structures, and more often related to runout, rather than landslides. This enables us to identify sectors at highest risk, as well as the process (e.g., runout vs. landslides) behind those risks (Figure 9). For the Kaligandaki area, we find that linear infrastructure is at much

greater risk than structures. 27 % of buildings are directly exposed to slope failures (Figure 9a, rightmost point on blue line), compared to 69 % of roads (Figure 9b, rightmost point on blue line). 27 % of buildings are exposed to runout (Figure 9 a, rightmost point on red line) while 65 % of roads are exposed to runout (Figure 9 a, rightmost point on red line). These numbers also indicate that many road segments are exposed to multiple risks from both slope failure and runout. Analyzing the statistical distribution of failure probabilities for structures (Figure 9 a) and roads (Figure 9 b) reveals that most structures are threatened by landslides/runout with a relatively low failure probability (<=10 %, stacked bars in Figure 9 and b). Nearly all exposure of buildings to damage from high probability (>50%) comes from runout, indicating that there are few buildings on unstable slopes with high failure probabilities, but such failure-prone slopes still contributes to downslope risk via runout.

#### 5.2 Leveraging LSO level data to inform management

As a last experiment, we deployed the LSO model to study the impact of vegetation and land management on landslide risk, deriving useful information for sustainable development and restoration planning in mountain regions. To demonstrate such a scenario analysis, we consider two common scenarios. The first scenario is complete deforestation, while the second scenario consists of reforestation of potentially unstable slopes (i.e., on pixels that are identified as LSO objects below the tree line). Note that neither of the two scenarios is designed as a realistic management plan for the Kaligandaki basin. The "deforestation" scenario could be used, for instance, for environmental accounting (such as in the United Nations Systems for Eco-Environmental Accounting, SEEA, Edens et al., (2022)) to understand where nature provides the greatest benefits for landslide prevention. The "reforestation" scenario could be used for selecting areas where reforestation could lead to greatest benefits for risk reduction. For both scenarios we consider that reforestation would increase root cohesion by 2kPa.

For buildings and roads, landuse changes could lead to a major change in exposure (Figure 9 c and d). For the reforestation scenario, the fraction of buildings exposed (considering on LSOs for clarity) would decrease from more than 30 % (rightmost point on blue line, Figure 9 c) to 7.5 % (rightmost point on green line, Figure 9 c) and the number of exposed road segments would decrease from 65 % to 38% (Figure 9 c). Deforestation only slightly increases the exposure buildings by around 1.5 % compared to the baseline. Reforestation could decrease the fraction of roads at risk from around 65 % to less than 40 %.

A spatial assessment indicates that response of risk to reforestation is mostly driven by the distribution of assets (Figure 10). The reforestation scenario targets LSOs and thus benefits structures and roads (blueish colors in Figure 10 a). For deforestation, there is a relatively small change in risk which is not because there is no change in landslide occurrence, but because only few buildings and roads are in forest areas (green in Figure 10). Instead, there is a clear increase in LSO occurrence and failure probabilities in deforested areas (see red colors in currently forested areas in Figure 10 b). Note increased/decreased risks from deforestation/reforestation can also impact areas outside where landcover is changing. These extended spatial impacts are because even if one cell of an LSO is located in a deforestation/reforestation area, the change in local failure probability would alter the average failure probability of the LSO, and of the associated runout pathway.

Figure 9: Statistical analysis highlights cumulative risk from landslides and runouts. a) and b) highlight which fraction of buildings (a) and roads (b) is exposed to landslides/runout with a certain probability. The total y value indicates the total fraction of exposed buildings/roads. Panels c and d show the cumulative exposure (from both landslides and runout) under two different scenarios, deforestation (red dotted line) and reforestation (green dash-dotted line), compared to the baseline (blue solid line).

Figure 10: Impacts of landuse on LSO failure probability. Shown is the difference in failure probability between a scenario where all current forest (green) is removed (a) and a scenario where reforestation occurs one currently unstable slopes (b). Runout pathways associated with LSOs are not shown for clarity. Road layer © OpenStreetMap contributors 2025. Distributed under the Open Data Commons Open Database License (ODbL) v1.0.

#### 6. Discussion

Landslides and mass movements are important means of natural erosion in mountain landscapes (Attal and Lavé, 2006; Lavé J. and Avouac J. P., 2001; Marc et al., 2019a). At the same time, landslides pose major hazards to livelihoods (Petley, 2012), either through direct exposure, or through their impacts on water infrastructure (Fort et al., 2010; Schwanghart et al., 2018) or linear infrastructure, such as powerlines and roads (Emberson et al., 2020; McAdoo et al., 2018). Climate and landuse change might increase landslide hazards (Gariano and Guzzetti, 2016), while non-strategic planning of infrastructure and other development can increase both hazards and exposure, leading to increased risks for human lives and infrastructure assets. A better understanding of landslide hazards under current and future conditions, as well as proactive planning for reducing such hazards can play an important role in designing landscape scale interventions (such as deploying nature-based solutions at scale, or for strategically siting assets in less hazard-prone areas), and for improving our understanding of geomorphologic processes on hillslopes and in river channels.

Herein, we argue that integrating approaches for susceptibility mapping with hillslope connectivity can improve the modeling and management of landslides in some critical ways. To operationalize this idea, we introduce the COHESION (COnnected HillslopE Susceptibility for slOpefailure and ruNout). Traditional susceptibility mapping approaches, while effective for identifying failure-prone areas on large scales and data-scarce regions, fall short in addressing the magnitude and interconnected nature of slope failure processes. The COHESION framework bridges this gap by incorporating connectivity into susceptibility mapping and improves large-scale assessments of landslide hazards. For instance, our findings indicate that connectivity significantly increases which areas area are classified as being at risk, as downslope runout can extend risks far beyond the initial failure zone. For instance, we find including runout into our analysis increases the area at risk by more than 60 %.

After calibration, we found that COHESION reproduces the statistical distribution of slope failures well (KS = 0.10). However, we also found that the modelled landslide density exceeds the value indicated by available remote sensing assessments. For instance, Marc et al.'s (2019a) assessment of several areas in Nepal, including Kaligandaki, indicated a density of unstable slopes of 200-250 m<sup>2</sup>/km<sup>2</sup>/year (0.02-0.025 %/year). Our modelled densities are two orders of magnitude higher (0.9%/year), yet still much smaller than other modeled estimates in Nepal (Kincey et al., 2024). With that regard, some aspects should be noted. Firstly, our handling of hydrology (as saturation along the hillslope flow paths) might overestimate the saturation, compared to approaches based on, e.g., transient water accumulation. Secondly, as we scaled  $\phi$  with  $\alpha$  we did not set a lower limit to  $\phi$ , which might result in unrealistically low values of  $\phi$ . E.g., if the slope angle in a cell was  $\alpha = 20^{\circ}$ , then would be  $\phi = 20^{\circ} * 0.6 = 12^{\circ}$ , very low compared to laboratory and observational data, typically placing  $\phi$  in the range of, e.g.,  $28^{\circ} - 45^{\circ}$  (Schellart, 2000) or 17 - 45° (Schmidt and Montgomery, 1995). Indeed, when setting a lower limit of 30° to  $\phi$ , we find that the unstable area decreases to 0.04 %. Lastly, some observational bias might also play a role, as it has been shown that extending the observed area results in much greater estimates of landslide densities (Harvey et al., 2025). Also other metrics of COHESION, such as threshold precipitation, are in a very similar range to observations (Dahal and Hasegawa, 2008).

Our model evaluation based on observed landslide occurrence highlights several avenues for future research as well as data needs. Firstly, having some information on observed landslides is critical to calibrate the model. As our handling of moisture is statistical, and not based on specific events or seasons, we needed to compare general statistics, such as angles of unstable slopes to observations, rather than actual landslide occurrence to observations. Herein, we focused on the slope angle to understand if modelled and observed landslides occur on slopes of similar steepness. Additional parameters could be

675

compared as well, such as the area of LSOs, but we found such a comparison difficult because of the relatively low resolution of our model (30m) compared to the area of observed landslides. Secondly, our 630 analysis highlights that COHESION is most sensitive to soil cohesion (c) and internal friction angle ( $\phi$ ) in terms of spatially distributed parameters, similar to studies deploying higher fidelity models (Almeida et al., 2017). We find that expressing  $\phi$  as a function of slope angle is needed for a well-fitting model, closely aligned with observations from large-scale landslide datasets (Emberson et al., 2022) and local studies for monsoon-induced landslide in Nepal (Burrows et al., 2023). Thirdly, data on soil depth and 635 thus the only distributed subsurface parameter used in this framework are of low resolution compared to other types of data (500 m, compared to 30 m for topography or landuse). With this regard, exploring further links between observed topography (slope angle), climate, and subsoil processes might be worthwhile, as studies suggest that e.g., slope angle (Prancevic et al., 2020) and long-term climatology (Marc et al., 2019b) influence the spatial distribution of subsoil properties.. Fourthly, soil saturation and 640 thus slope failures (and threshold precipitation) are sensitive to antecedent conditions (Gabet et al., 2004). Our model, being statistical, is instead based on a representative average wet season, thus modeling different representative seasons (e.g., with high vs. low average precipitation rates) could allow to estimate ranges of failure probabilities for each LSO.

Coupled assessments of landslide susceptibility and connectivity can inform scenarios and planning 645 irrespective of the type of deployed susceptibility mapping. Herein, COHESION uses a probabilistic approach to identify conditionally unstable slopes and their failure probability. Such information on failure probability can be useful for economic evaluations (e.g., by multiplying the value of a structure with the probability of it being damaged), or for estimating the average annual contribution of landslides to a basin's sediment budget (Vogl et al., 2019a). However, susceptibility maps as input into COHESION 650 (as input into Step 4.4 in Figure 2) can be generated through other common approaches, e.g., multicriteria analyses (Asadi et al., 2022; Kavzoglu et al., 2014; Lorentz et al., 2016), factor of safety assessments based on single empirical rainfall thresholds (Dahal and Hasegawa, 2008; Gabet et al., 2004), or assessments of co-seismic hazard (Milledge et al., 2019). Through frameworks for rapid terrain analysis (Schwanghart and Kuhn, 2010), global digital elevation models with a high resolution 655 (OpenTopography, 2025), and increasing computation resources, the proposed ex-post connectivity assessment will be feasible on any scale on which susceptibility assessments are performed today. The resulting additional information on the magnitude of potential slope failures and the risk of runout can then be used to inform infrastructure planning and to determine optimal locations for deploying traditional (grey), green, or grey-green engineering measures to protect existing assets.

COHESION also offers avenues to link hillslope-channel connectivity to fluvial processes and water infrastructure. For instance, information on runout and landslide volume could be intersected with river networks and be used as inputs into network scale sediment connectivity models, such as CASCADE such as CASCADE (Schmitt et al., 2016) or others (Czuba, 2018; Czuba et al., 2017; Czuba and Foufoula-Georgiou, 2014). Such a coupling would not only be interesting from the perspective of 665 developing a catchment perspective on connectivity including catchment and river processes, but could also be of operational use for downstream water infrastructure. Many dams in Nepal are at risk from landslides(Fort et al., 2010; Sangroula, 2009; Schwanghart et al., 2018; Sharma and Awal, 2013), not only endangering their operation but also downstream lives, as landslides can trigger dam failures. To our knowledge, this hazard is not commonly included in decisions about dam development, a major limitation 670 in a hydropower-rich country. In this context, COHESION could help to develop assessments of upstream risk for hydropower plants beyond common local geotechnical assessments of slope stability around the dam site.

Understanding landslides is critical for understanding landscape sediment connectivity, fluvial sediment transport and for managing disaster risk for human livelihoods and infrastructures in mountain regions. Despite prevalent data limitations, our COHESION framework highlights that combining susceptibility mapping and connectivity assessments is a promising landscape-scale approach in modelling landslide

hazards in a scalable, spatially explicit way. This apporoach supports science, and helps to design adaptation and mitigation strategies against landslide hazards in mountain ranges worldwide.

#### 7. Acknowledgement

This research was supported World Bank contract 7184416 "Mapping and Valuing Ecosystems Services, and Prioritizing Investments in Select Watersheds in Nepal and Pakistan to support Sustainable Hydropower". The authors acknowledge the contribution of Dr. David Simpson (American University) for many productive discussions regarding the treatment of landslide probabilities as prerequisite for economic damage analysis and the guidance of Dr. Urvashi Narain (World Bank). ChatGPT 40 was used to review and harmonize some code for ex-post analysis of results to improve readability, resulting changes were carefully reviewed by the first author.

#### 8. Author contributions

RJPS: Conceptualization of the COHESION Framework, software, methodology, formal analysis (equal), writing – original draft, writing – review and editing (equal), visualization, supervision, design of experiments. SB: formal analysis (equal), software, methodology, writing – original draft, writing – review and editing (equal), visualization, data-curation. AV: conceptualization, funding acquisition, writing – review and editing. OM: methodology, writing – review and editing.

#### 9. Data availability

The model code for the COHESION model and for model verification are available on Zenodo https://zenodo.org/records/16595396.

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
