# Peer review of "Leveraging hillslope connectivity for improved large-scale assessments of landslide risk"

_EGUsphere, 2025_

## Referee Comment (RC1)

**General comment**

The paper proposes a probabilistic framework for landslide risk assessment in large areas, which couples landslide susceptibility with hillslope connectivity, so to better estimate landslide volumes and runouts, which allows a more reliable identification of assets at risk. The topic is of high interest for the readership of NHESS. The manuscript is well structured, and the English language is good, so that the reading is easy.

Although the idea of linking hillslope connectivity with landslide susceptibility is valuable, the manuscript suffers from some methodological flaws that make the present version not suitable for publication. Specifically, the major issues deal with the probabilistic modelling of rainfall; the formulations used for the assessment of slope stability and the calibration of the parameters therein; the choice of treating only rainfall with a probabilistic approach, while other parameters which affect the results even more than rainfall are also uncertain but have been assumed deterministically known; how runout has been evaluated; the analysis of the assessed risk, which omits details seemingly important without explanation.

Some of the above listed major issues pose doubts about the quantitative results presented and the conclusions drawn. Besides, there are several minor points that should be improved. Both major and minor issues are explained in detail hereinafter.

**Detailed comments**

Line 68 – "Numerical" should be "mathematical", as the term numerical refers to techniques to solve the (PD) equations of mathematical models.

Line 100-101 – Susceptibility maps refer to the proneness of a slope to failure. Thus, it is not surprising that they don't consider connectivity and runout. I totally agree with the authors, but the point here should be that mapping susceptibility is not enough to assess landslide risk, rather than stating that susceptibility does not consider exposure.

Line 141 – The second "(B)" should be "(C)".

Line 180 – The equation adopted for calculating FS is an approximation, as (to my understanding) the soil column of thickness  $z_i$  is schematized as being totally saturated in the lower part of thickness  $m_i * z_i$  and totally dry above. I suggest specifying the hypotheses behind the chosen formulation, especially because the proposed methodology could be applied also with FS calculated with different equations (still with the infinite slope assumption, but with a formulation closer to reality that considers the presence of unsaturated soil). This would help highlight the general validity of the proposed approach.

Line 184 – It would be preferable to define  $z_i$  as soil thickness, and to specify that FS is evaluated assuming that the slip surface is at the base of the soil column of thickness  $z_i$ .

Lines 185-186 – I understand that you are considering that the soil has always the same unit weight (i.e., 15.7 N/m³, corresponding to about 40% soil porosity if the density of solid particles

is assumed around 2700 kg/m³), even in a catchment as large as 6000 km². This assumption looks doubtful, and if the absence of the subscript is not a typo, it should be motivated somehow based on soil characteristics.

Line 187 – In the equation, the friction angle is without the subscript i, as it was constant. I understand that this is a typo. Maybe also  $\gamma_s$  should be  $\gamma_{si}$ ?

Line 193 – The assumption of linking friction angle to slope inclination relies on considerations made in the reference given at line 197. However, in that paper slope stability modelling was not the focus, but just a means to understand what mechanism might be behind observed landslides. Using a relationship like equation 2 for slope stability assessment (even at regional scale) deserves more discussion. From a physical point of view, it makes sense to expect that stronger soil rests on steeper slopes (although cohesion may change this simplistic picture). In any case, this does not imply that soil resting on gentle slopes must be weak, as equation 2 implies (there is no physical reason for this). In any case, even on steep slopes, assuming that friction angle must always be (much) smaller than inclination angle gives the cohesion (both soil effective cohesion and additional cohesion from root reinforcement) the entire task of ensuring slope equilibrium.

I played a bit with equation 1. For instance, evaluating FS at the depth z=2 m and with the values of soil strength parameters obtained through calibration (i.e., b=0.65 and c=5 kPa) indicates that, without roots, the equilibrium of all slopes with  $\alpha$ >28°, even if completely dry (mi=0), is impossible. If we consider a 50% saturated slope (mi=0.5) all slopes without roots with  $\alpha$ >17° would fail. Root reinforcement, introduced as additional 2 kPa cohesion, totally changes the picture: all slopes up to  $\alpha$ =18° are unconditionally stable, even if the soil is totally saturated, and with mi=0.5 failure occurs for any  $\alpha$ >26°. In short, slopes are too weak.

Considering a 30° inclined slope with z=2m and  $m_i$ =0.5 (FS=1.01), reducing cohesion from the value with roots, 7 kPa, to 6 kPa makes FS reduce of about 8.2% (FS=0.93). To get the same reduction of FS,  $m_i$  should grow up to 0.71 (more than additional 170 mm infiltrated rainwater). Thus, slope stability seems less sensitive to the effects of rain than to (arbitrarily) chosen soil strength parameters.

I understand that the study refers to an area where information about soil properties is uneasy to obtain, and that the proposed methodological framework would remain valid even with a weak modelling of soil stability. But the results are so much affected by the way soil strength is modelled (think about reforestation/deforestation effects, in a model where the additional cohesion due to root reinforcement is by far the major control of slope stability), that I wonder how many of the obtained quantitative results would survive if just a little change (improvement) of slope stability modelling was made.

Line 207 – The subscript i is missing at the left-hand side. The DEM resolution is constant throughout the entire map of the basin, so B should not have the subscript i.

Line 211 – Is K constant everywhere? Or should it be Ki?

Line 218 – It would be clearer if the equality  $Q_i=q_i*A_{D,i}$  was given in equation 3 (as it is in equation 12 at line 255). To my understanding, B (the size of a pixel) should be constant so without the subscript j.

Line 220-221 – The meaning of kfi and how it is estimated should be described.

Line 225 – There is a typo. I guess it should read " $Q_{R,j}$  and  $Q_{D,j}$ ".

Line 227 – The word "runoff" is probably missing at the end of the line.

Lines 254-257 – The width of a pixel was indicated with capital B (here it is b), and it should not have the subscript i, as the DEM resolution is not changing from pixel to pixel.

Line 257 – The equation is incorrect ( $Q^*$  should be only at the right-hand side).

Lines 271-284 – In equation 15, p\*i(e) represents the precipitation capable of triggering a landslide at cell i depth during an event e, and rainfall events may have any duration. However, the EV type I (Gumbel) distribution, and the parameters therein, should be calibrated based on a set of measurements of extreme rainfall all sharing the same duration (e.g., Koutsoyiannis et al., 1998). For instance,  $\mu_i$  and  $\sigma_i$  would be different if the probability distribution refers to hourly rainfall depths or daily rainfall depths (as well as for any other different rainfall duration that might be considered). What kind of rainfall data (hourly? daily?) has been used to estimate the parameters of equations 16 and 17 at the positions of the rain gauges represented in Fig. 3, then used to interpolate them throughout the whole basin area with kriging? This should be specified and, even more importantly, it should be explained how the same probability distribution (obtained from a set of rainfall data with a given duration) can be used to describe the probability of rainfall depths falling during events with different durations. To avoid mistakes in the calculated probabilities, maps like those shown in Figure 3 should be produced for any possible rainfall duration, and the appropriate one should be chosen according to event duration, so to get the correct parameters (and to calculate the correct probability). Alternatively, a fixed rainfall duration might be assumed, but in such a case it would not be correct to refer to rain events (e.g., if the fixed duration is one day, as it may seem from the text at lines 491-495, p\* should be defined as the critical daily precipitation rather than event precipitation).

Lines 290-293 – Again, case-specific information is mixed with the general methodology.

Lines 294-300 – To my understanding, all the spatially variable parameters except precipitation have been considered deterministic. This should be better underlined, as the sentence at lines 299-300 may be misleading, and the sentence at the beginning of section 4.3 does not help (lines 241-242: "Herein, we consider for the spatiotemporal variability in soil moisture through a statistical approach"). Indeed, the variability of soil properties suffers the same uncertainty as rainfall, meaning that soil properties are available at a very small number of (sparse) locations, compared to the extension of the basin where the analysis is run. And I believe that also soil thickness  $z_i$  at each node of the grid is a mere estimate. The uncertainty affecting soil and slope properties can be treated with a probabilistic approach in a similar way to what you

do with rainfall, although with a different probability distribution (e.g., Roman Quintero et al., 2025). Given the poor results obtained in terms of identification of unstable zones (lines 608-610), likely due to the way soil strength parameters have been assigned (see some of my comments, but also your discussion at lines 614-618), probably treating also the uncertainty of soil parameters with a probabilistic approach (e.g., Almeida et al., 2017; Roman Quintero et al., 2025) would be worth.

Lines 304-305 – The last words in the figure caption seem to be a typo and should be removed (i.e. "Connected slope failure assessments").

Line 306 – From this line onward, the part of the methodology dealing with connectivity and landslide objects is described. This is an essential part of the proposed methodology and has nothing to do with how the probability of failure is calculated; thus, it would be better to start here a separate subsection 4.4.

Line 329 – More details about how the function works and how it has been used (it has a parameter to be set) should be given.

Line 342 – There is a typo: a "T" is missing at the beginning of the line (it should be "This").

Line 367 – The statement is physically not correct: landslide volume determines the mass, but to calculate the kinetic energy also the velocity is needed, which in turn depends directly on the elevation drop from the unstable zone to the toe where the kinetic energy is evaluated (e.g., the available potential energy) and only indirectly on the slope gradient (owing to energy loss along a longer pathway). Indeed, empirical equation 24 considers only the elevation drop and neglects the gradient.

Lines 373-376 – It is unclear how the runout distance is considered: does it start from the toe of the slope, or does it also include the path travelled along the slope? In this respect, the statement "These two quantities are not independent, as longer runout distances naturally involve greater vertical drops" sounds at the same time obvious (the dependence is in equation 24) and misleading (do the authors here mean anything different from the link empirically expressed by equation 24?).

Lines 377-380. The description of the procedure is not clear, and I have doubts about its correctness (at least, from what I could understand). If the remaining runout distance is reevaluated with equation 24 at each downslope step with a new elevation drop, this implies that the already attained velocity of the mass sliding from upslope is neglected, as equation 24 refers to the position of the landslide scarp, where the mass is initially still. The sketch in the corner of Fig. 5 does not help: for the  $\gamma_4$  path (which consists of three pixels), a drop  $\delta_h$  is represented (it should be  $\delta H_h$ , if I correctly understand), which should be the "the elevation drop from LSOk to i" (as explained at lines 382-383). To improve the clarity, besides correcting all the inaccuracies in the symbols and in the figure, I suggest indicating the position of node i (it should be where the vertical segment marking  $\delta_h$  touches the ground line). However, the

corresponding horizontal segment should be  $\delta L_h$  (explanation at line 383: " $\delta L_h$  is the horizontal distance from LSOk to I") and not L4, as indicated in the figure.

Lines 377 and 379 – Equation 30 does not exist. Please correct.

Line 380 – Please, clarify what you mean with "channel network". The nearest river branch?

It seems that Figure 5 has typos, as the runout path of LSO4 is indicated as  $\gamma_3$  and the exponent of V is missing in the equation.

Lines 410-412 – The KS test is usually carried out comparing the KS statistics with a threshold value, which depends on the number of elements, m and n, belonging to the two compared sets  $(KS_{lim} = \frac{1.63}{m \times n})$  and allows calculating the p-value of the test to evaluate its statistical significance. I understand that here you are not really trying to check if predicted and observed failures share the same distribution of slope inclinations, but you have m=1170 observed landslides and n, the number of modelled LSO, which is not given in the paper, but should be very high looking at the smoothness of the cumulative blue curves of Figure 6c and 6d (and to the too large extension of unstable areas discussed at lines 609-610). Assuming for instance n=10000, it results  $KS_{lim} = 0.014$ , meaning that observed and modelled failed slopes are very differently distributed, even when the minimum value KS = 0.1 is obtained through calibration. I think that this bad performance, which does not hamper the validity of the proposed approach (a better slope stability modelling could be implemented within the same framework), relies on the friction angles used for assessing slope stability, i.e. the inadequacy of equation 2, the relationship linking friction angle to slope inclination (as indeed you argue in the discussion). Correcting that relationship would be easy, so to get better results.

Line 413-414 – Soil specific weight  $\gamma_s$  has been used in the FS equation. Although soil density and specific weight are obviously related, it would be better to run the sensitivity analysis with reference to the symbols and quantities that appear in the equations used to calculate FS.

Table 2 – The last row of the table reports the friction angle as a calibration parameter, while it is calculated through the empirical relationship given in equation 2, and indeed the parameter b of that equation is indicated as calibration parameter. I suggest removing the last row of the table, as this might appear inconsistent.

Figure 6 – The black arrows plotted above a dark blue are hardly visible. I suggest changing the color of the arrows to a lighter one.

Lines 510-513 – The volume of a landslide may not only reduce while it moves downslope, but it may also increase, due to entrainment of more soil and debris along the path. I suggest rephrasing this sentence to consider also this instance.

Lines 526-529 – Please, double check the references to the curves and the panels of figure 9 for evaluating the percentage of exposure of roads and buildings to landslide and runout, as it seems that there are some discrepancies.

Lines 531 and 532 – So far, the word "building" was used, while now it is "structures". To avoid misunderstanding, I suggest keeping the same word.

Lines 530-535 – The description is unclear. From the graphs of figure 9a and 9b a clear different behavior of building and roads as regards LSO and runout exposure appears, which is not clearly discussed in the text (i.e., buildings are more exposed to LSO with small probability and uniformly exposed to runout with any probability, while the opposite holds for roads). Also, the English language should be improved here.

Line 549 – I guess there is a typo in the text in the parentheses, as "on" should probably become "only". This is confirmed by figures 9c and 9d, where the effects of deforestation/reforestation have been evaluated only for LSO exposure. However, this choice should be better motivated (see one of my following comments): what does it mean "for clarity"? Are the effects on runout exposure less clear? And why?

Lines 550-554 – I think that the curves plotted in fig. 9c and 9d deserve a deeper comment. I see that deforestation not only affects very slightly the total % of exposed buildings (as noted by the authors), but it even reduces the buildings exposed to failure probabilities smaller than 0.5. From the cumulative curves, it is difficult to understand what is behind this result: a histogram could allow the reader to better understand if, and to what extent, the buildings exposed to small probability of failure (<0.5) have moved to higher probabilities. This effect looks clearer for the exposure of roads, where clearly deforestation moves assets from low-probability exposure to p>0.9, as indicated by the increase of the slope of the cumulative % in the interval (0.9, 1.0).

Lines 555-564 – Here the reason behind the different effects of deforestation and reforestation: reforestation is made where some landslide risk was present, so it affects failure probability exactly where reducing failure probability is needed. Differently, deforestation occurs where forests, and hence neither buildings nor roads were present, and there is a slight indirect effect only because pixels belonging to the same LSO affect the mean value of the failure probability of that LSO. This makes sense, but it underlines that probably the effects of deforestation on risk should be evaluated on the exposure to the runout rather than on LSO. Why was this choice made?

Figure 9c – Why the green curve ends at p=0.9? It is a cumulative curve, so it should cover the entire probability interval (up to p=1). Even more strangely, the orange dashed curve goes beyond P=1, which is meaningless. Please, correct.

Line 578 - A typo: "occurs on" instead of "occurs one".

Lines 614-618 – This paragraph would better fit in the introduction than in the is a repetition of what already discussed in the introduction, and it has

Lines 630-634 – This statement ("well-fitting model") is not supported by the results, and the sensitivity to soil strength parameters confirms that their estimation is the major weak point of this study, that would deserve a deeper discussion.

**References**

Almeida, S., Holcombe, E. A., Pianosi, F., and Wagener, T.: Dealing with deep uncertainties in landslide modelling for disaster risk reduction under climate change, Nat. Haz. Earth Syst. Sci., 700, 225–241, https://doi.org/10.5194/nhess-17-225-2017, 2017.

Koutsoyiannis, D., Kozonis, D., and Manetas, A.: A mathematical framework for studying rainfall intensity-duration-frequency relationships, J. Hydrol., 206(1-2), 118-135, https://doi.org/10.1016/S0022-1694(98)00097-3, 1998.

Roman Quintero, D. C., Marino, P., Abdullah, A., Santonastaso, G. F., and Greco, R.: Large-scale assessment of rainfall-induced landslide hazard based on hydrometeorological information: application to Partenio Massif (Italy), Nat. Hazards Earth Syst. Sci., 25, 2679–2698, https://doi.org/10.5194/nhess-25-2679-2025, 2025.